# Diabetic Retinopathy: The Role of Mitochondria in the Neural Retina and Microvascular Disease

**DOI:** 10.3390/antiox9100905

**Published:** 2020-09-23

**Authors:** David J. Miller, M. Ariel Cascio, Mariana G. Rosca

**Affiliations:** Department of Foundational Sciences, Central Michigan University College of Medicine, Mount Pleasant, MI 48858, USA; mille15d@cmich.edu (D.J.M.); ariel.cascio@cmich.edu (M.A.C.)

**Keywords:** diabetic retinopathy, mitochondria, oxidative stress, redox, photoreceptor

## Abstract

Diabetic retinopathy (DR), a common chronic complication of diabetes mellitus and the leading cause of vision loss in the working-age population, is clinically defined as a microvascular disease that involves damage of the retinal capillaries with secondary visual impairment. While its clinical diagnosis is based on vascular pathology, DR is associated with early abnormalities in the electroretinogram, indicating alterations of the neural retina and impaired visual signaling. The pathogenesis of DR is complex and likely involves the simultaneous dysregulation of multiple metabolic and signaling pathways through the retinal neurovascular unit. There is evidence that microvascular disease in DR is caused in part by altered energetic metabolism in the neural retina and specifically from signals originating in the photoreceptors. In this review, we discuss the main pathogenic mechanisms that link alterations in neural retina bioenergetics with vascular regression in DR. We focus specifically on the recent developments related to alterations in mitochondrial metabolism including energetic substrate selection, mitochondrial function, oxidation-reduction (redox) imbalance, and oxidative stress, and critically discuss the mechanisms of these changes and their consequences on retinal function. We also acknowledge implications for emerging therapeutic approaches and future research directions to find novel mitochondria-targeted therapeutic strategies to correct bioenergetics in diabetes. We conclude that retinal bioenergetics is affected in the early stages of diabetes with consequences beyond changes in ATP content, and that maintaining mitochondrial integrity may alleviate retinal disease.

## 1. Introduction

Diabetes mellitus is a growing public health problem, reaching pandemic proportions in the United States and worldwide [1]. Diabetic retinopathy (DR) is the leading cause of irreversible visual impairment and blindness in the working-age population [2]. The Diabetes Control and Complications Trial concluded that tight metabolic control can delay the development and slow the progression of DR. However, good metabolic control is often difficult to achieve and does not guarantee complete protection against DR, suggesting that there are additional contributing factors that remain to be discovered [3,4]. While targeted therapies are effective in mitigating the sight-threatening complications of proliferative diabetic retinopathy (PDR) [5], new therapeutic approaches are needed to manage the milder non-proliferative disease. Thus, there is an urgent need to better understand the early stages of DR in order to develop new strategies to halt its progression.

DR is clinically defined as a microvascular disease [6], and can be broadly classified into two distinct stages on the basis of the presence of neovascularization. While non-proliferative diabetic retinopathy (NPDR) is characterized by blood flow alterations, pericyte loss, downregulation of endothelial cell tight junctions [7], and thickening of the basement membrane [8], PDR presents with sight-threatening neovascularization that may precipitate retinal detachment and blindness. Recent work has shown that retinal neurodegeneration precedes clinically detectable microvascular damage [9,10,11,12,13]. Since Wolter’s first observation of neuronal cell death in the diabetic retina [14], numerous studies have described early neuronal apoptosis and alterations in visual signaling. Retinal ganglion cells (RGCs) of the optic nerve undergo apoptosis at a rate higher than any other retinal cell [15]. These changes are associated with a subjective decline in the quality of vision including impaired contrast sensitivity and color vision [16,17,18], and altered visual signaling as assessed by the electroretinogram (ERG). In addition to changes in the a- and b-waves on the ERG, alterations in the amplitude of the oscillatory potential (photopic and scotopic oscillatory potentials, which are initiated in the inner retina [19]) have been suggested to predict the progression of DR [20,21].

In light of these findings, new discoveries into retinal physiology have emphasized the role of the neurovascular unit in DR [6], which refers to the physical and biochemical interaction between neurons (RGCs, amacrine cells, bipolar cells, and horizontal cells), glia (Müller cells and astrocytes), and the microvascular network (endothelial cells and pericytes) [22,23]. The key role of this interaction in neurodevelopment [24] and normal neurovascular signaling [25] has led to the hypothesis that DR may result from the uncoupling of the neurovascular unit [26,27]. Nevertheless, the effect and timing of cellular dysfunction throughout the neurovascular unit in DR has yet to be determined.

One of the classical and prevailing theories explaining the pathogenesis of DR is that diabetes enhances oxidative stress, which in turn damages the retinal microvasculature [28]. The term oxidative stress refers to an imbalance between reactive oxygen species (ROS) production and antioxidant defenses. Because of their role in oxidative metabolism, mitochondria are key sources of increased ROS in diabetes [29,30,31]. Oxidative stress originating in mitochondria of endothelial cells has been reported to enhance multiple seemingly independent pathways, each contributing to the development of microvascular complications [32,33]. Most current knowledge is derived from this “unifying theory” that was developed on cultured aortic endothelial cells and has since been extrapolated to the retinal microvasculature. However, recent work by Du et al. [34] determined that diabetes-induced oxidative stress originates from the photoreceptors rather than endothelial cells. In this model, photoreceptor-induced oxidative stress was associated with increased inflammation, which is widely regarded as an important pathogenic mechanism of DR, and contributes to vascular regression in the diabetic retina [35]. The critical role of the neural retina in the development of microvascular disease is further supported by studies of patients with retinitis pigmentosa who exhibit both photoreceptor degeneration and protection against DR [36,37].

While performing the core metabolic function of energy production, mitochondria are critical gears in a currently expanding number of cellular functions including redox homeostasis [38] and programmed cell death [39]. An increased mitochondrial oxidative stress reveals a change in mitochondrial function. The importance of understanding the role of bioenergetics in the diabetic neural retina is supported by the knowledge that inherited mitochondrial diseases cause retinal disease and visual impairment [40], and is further highlighted by the heterogeneity of the neural retinal cells regarding the contribution of their mitochondria to cellular ATP and oxidative stress [41,42]. This review will summarize the recent developments related to alterations in mitochondrial bioenergetics in the neural retina, as well as the consequences of these alterations on retinal function. We will conclude by acknowledging emerging therapeutic approaches to correct mitochondrial bioenergetic-related functions and maintain the mitochondrial integrity in diabetes.

## 2. Normal Retinal Structure

The retina is a highly organized tissue consisting of at least 10 distinct layers, which can be broadly divided into an inner and outer retina (Figure 1).

The inner retina includes the RGCs as well as two nuclear layers with the photoreceptor soma. Photoreceptors are the light-sensitive cells responsible for phototransduction and present as either rods and cones expressing the visual pigments rhodopsin and opsin, respectively. The outer retina includes the photoreceptor outer segments (OSs) and the underlying retinal pigment epithelium (RPE). The RPE rests on Bruch’s membrane, a multi-layered structure that separates the outer retina from the choroid choriocapillaris. The inner retina receives blood from three local vascular plexuses, while photoreceptors are primarily supplied by the choriocapillaris. Therefore, although the photoreceptors are physically distant from the inner retina where DR manifests as a microvascular disease, both structures contribute to the pathophysiology of DR but their cooperative signals are yet to be identified.

## 3. Pathophysiology of DR

The pathogenesis of the early stages of DR remains poorly understood. Pericyte death has been considered the central mechanism for the loss of retinal vascular integrity in diabetes [43]. However, the seminal work of Mizutani et al. [44] revealed early and accelerated death of both retinal pericyte and endothelial cells in diabetic rodents and humans. While endothelial cells are replaced by proliferation, migration, or neighbor cell redeployment, pericytes do not regenerate, and their absence is evidenced by the presence of “pericyte ghosts” in the capillary wall. Dynamic high resolution microscopy determined that the decrease in blood flow favors the process of vasoregression [45,46]. As an indisputable event in the diabetic retina, pericyte loss has been observed in all rodent models of both type 1 (T1D) and type 2 (T2D) diabetes [47,48,49]. Moreover, genetic pericyte elimination recapitulates the early features of experimental DR, including acellular capillaries, microaneurysms, and blood–retinal barrier abnormalities, all of which underline the seminal role of pericytes to maintain retinal capillary integrity [50]. While pericytes likely play a similar role in humans [51], progress in this area is limited by the scarcity of human retinal tissue and the inherent difficulties of translational research [52].

Previous studies have focused primarily on the retinal microvasculature. However, a recent growing body of literature indicates that diabetes causes cellular dysfunction and loss of virtually all retinal cell populations [13,53,54,55,56,57,58], as measured qualitatively by ERG and quantitatively by optical coherence tomography, revealing a decrease in retinal thickness [10,59]. Diabetes-induced alterations of neuronal cells and photoreceptors are particularly important as the death of these cells is not matched by similar rates of regeneration [60]. Due to the large surface area of outer segments (OS), photoreceptors are highly sensitive to incident photons and have a high capacity for ion exchange that must be supported by ATP. Abnormalities in photoreceptors have been reported in multiple models of insulin-dependent diabetes in both rodents [61,62] and rabbits [63]. Similar observations have been reported in zebrafish exposed to hyperglycemia [64]. In diabetic patients, photoreceptor integrity is altered and the OS length shortened, changes that have been associated with decreased visual acuity [65,66,67]. While altered photoreceptor morphology appears modest at 3–6 months of hyperglycemia [62,68], the functional abnormalities are more severe and include impaired function of the Na^+^/K^+^ ATPase pump [69,70]. In photoreceptors, the Na^+^/K^+^ ATPase pump is critical not only for normal ion homeostasis, but also for the “dark current”, a physiologic event that can be assessed by the a-wave on the ERG. Subsequent studies have expanded upon this work and observed changes in the amplitude and latency of the a-wave as early events in streptozotocin (STZ)-induced diabetes [71]. Similar abnormalities in the ERG have also been noted in diabetic patients, and suggested to precede and predict the microvascular histopathology [72]. These findings are consistent with the hypothesis that early visual dysfunction precedes morphologic neurodegeneration and vascular regression in DR (Figure 2).

## 4. Retinal Bioenergetics and Mitochondrial Substrate Selection

### 4.1. ATP-Consuming Processes in the Retina

While all retinal cells rely on ATP as a fuel source, the photoreceptors are the largest consumers. Photoreceptors use more than 75% of oxygen of the retina and contain more than 75% of retinal mitochondria to produce large amounts of ATP by oxidative phosphorylation (Oxphos), which is necessary for phototransduction [73]. Phototransduction, the process by which photons are converted into electrical signals in photoreceptors, relies on the cycling of 11-*cis* retinal, a vitamin A derivative bound to an opsin G-protein-coupled receptor (GPCR). In the presence of light, 11-*cis* retinal is isomerized to all-*trans* retinal. This photoisomerization results in a conformational change of the opsin GPCR, leading to a signaling cascade that causes the closure of sodium ion channels, hyperpolarization of the cell, and decreased glutamate release with depolarization of bipolar cells initiating phototransduction. In the dark, 11-*cis* retinal holds the opsin GPCR in an inactive conformation allowing the entry of sodium ions with glutamate release, thus inhibiting bipolar cells. This latter process is referred to as the “dark current”, a high ATP consuming process needed to maintain a steady influx of sodium ions and keep a constant membrane potential.

In order to provide a constant supply of 11-*cis* retinal, all-*trans* retinal must be converted back to 11-*cis* retinal through a series of redox reactions collectively referred to as the visual cycle [74]. The visual cycle involves proteolysis of the visual pigment (opsin or rhodopsin) and release of all-*trans* retinal into the RPE, where it is converted to 11-*cis* retinal. The rate of 11-*cis* retinal regeneration is determined by the availability of ATP and nicotinamide adenine dinucleotide phosphate (NADPH) [75], further supporting the proposition that the visual cycle is highly dependent on bioenergetic support. Photoreceptors undergo daily shedding, losing approximately 10% of their OS to phagocytosis by the RPE [76]. Continuous shedding of “used” OS discs and replacement with newly assembled discs, a critical process to maintain normal photoreceptor function, also consumes a large amount of ATP and NADPH. Photoreceptors are supported by adjacent Müller cells [77]; studies have shown that disruption of Müller cell metabolism results in impaired assembly of nascent photoreceptor OS [78]. Thus, the retina is a highly active tissue and requires a remarkable amount of oxygen and ATP to sustain its normal functions.

### 4.2. ATP-Generating Processes in the Retina and the Heterogeneity of Retinal Bioenergetics

The major sources of ATP in the retina are extramitochondrial glycolysis and mitochondrial Oxphos (Figure 3). In the 1920s, Warburg and Krebs reported that the mammalian retina, as a whole, has a metabolism largely based on aerobic glycolysis, converting 80–96% of glucose to lactic acid [79]. However, more recent research has demonstrated that the distribution of glycolysis and oxidative metabolism varies throughout the retina [80]. While neurotransmission in the inner retina is supported almost entirely by glycolysis, phototransduction in the outer retina is supported by mitochondrial Oxphos [80]. Mitochondrial Oxphos occurs in the inner mitochondrial membrane in which invaginations called cristae greatly increase the surface area for electron transport and ATP production. The electron transport chain (ETC) consists of four complexes (I-IV) that oxidize nicotinamide adenine dinucleotide (NADH) and flavin adenine dinucleotide (FADH_2_) to NAD^+^ and FAD^+^, respectively. Through a series of redox reactions, the ETC transfers electrons towards molecular oxygen and H^+^ into the intermembrane space. This process creates a transmembrane electrochemical gradient, which is used by ATP synthase (complex V) for the phosphorylation of adenosine diphosphate (ADP) to ATP. In addition to the ETC complexes, mitochondrial Oxphos also relies on ubiquinone (coenzyme Q) and cytochrome c (cyt c), two mobile electron carriers that shuttle electrons between ETC complexes [81].

A comprehensive investigation into oxidative metabolism revealed that retinal mitochondrial Oxphos operates in basal conditions at maximal capacity without a significant reserve capacity [82], suggesting that mitochondrial defects have a significant impact on retinal energy homeostasis. In most tissues, Oxphos is a tightly coupled process in which substrate oxidation is paired by ATP synthesis [81]. In the retina, mitochondria are reportedly less coupled, allowing proton leakage through the inner membrane without ATP synthesis [82]. Weak coupling between electron transport and ATP synthesis suggests that mitochondrial oxidative metabolism in the retina supports other functions in addition to ATP production, such as maintaining the NADH/NAD^+^ and FADH_2_/FAD^+^ redox ratios. This concept is one of the core focus of our review, and will be further detailed in the following sections.

Despite the high demand for ATP and NADPH described previously, photoreceptor OSs have limited glycolytic capacity and are devoid of mitochondria (Figure 1), relying on the inner segments (IS) for their energetic needs. Accordingly, photoreceptor ISs have the highest capacity for glycolysis, tricarboxylic acid (TCA) cycle, mitochondrial Oxphos, and creatine phosphate-mediated shuttling of ATP into the cytosol [80]. Photoreceptor ISs contain high amounts of hexokinase 2 (HK2) on the mitochondrial outer membrane, which catalyzes the rate-limiting step of glycolysis, namely, the conversion of glucose into glucose-6-phosphate. From here, a portion of glucose proceeds through glycolysis and the TCA cycle, which provides the GTP needed for phototransduction. Glucose-6-phosphate is also utilized in the pentose phosphate pathway, which is the primary source of NADPH used in anabolic reactions and the regeneration of cytosolic reduced glutathione (GSH), a major antioxidant defense mechanism. Between the two photoreceptor populations, cones contain 10-fold more mitochondria and thus have a much greater ATP-generating capacity than rods. Cone ISs also contain greater amounts of creatine phosphate, suggesting that cones provide the cytosolic ATP more efficiently than rods in the setting of high energetic demand [80].

The outer retina exhibits light-induced changes in oxygen consumption and ATP production [82]. As mentioned previously, phototransduction requires a steady influx of ATP and NAPDH both for the regeneration of 11-*cis* retinal and photoreceptor OS. Light has been shown to stimulate the accumulation of ribose-5-phosphate, an intermediate in the pentose phosphate pathway, which likely reflects increased NADPH production and anabolic metabolism. Oxygen consumption also correlates with the rod dark current, which imposes a high energetic demand in mammals [83], accounting for 41% of total retinal oxygen consumption [84]. When oxygen supply is inadequate, the dark current may be partially supported by glycolysis, indicating that more ATP is needed and extracted in the dark from energetic fuel substrates through their oxidation.

In contrast to the outer retina, the inner retina does not exhibit significant light-induced changes in oxidative metabolism. Nevertheless, neurons and glia of the inner retina also require a steady state [ATP] for neurotransmission. Müller cells, the most abundant glial cell in the retina, are critical to the maintenance of the neurovascular unit and perform important functions such as synaptic transmission regulation, handling of nutrients and waste products, maintenance of the “tightness” of the blood-retinal barrier, and survival of neurons and endothelial cells. Müller cells rely primarily on glycolysis and are rich in glycogen reserves [85,86]. Although glucose is their preferred substrate, Müller cells also utilize extracellular glutamate [87], and for this reason are believed to play a role in preventing glutamate excitotoxicity. Cell culture experiments have confirmed that Müller cells exhibit aerobic glycolysis and provide lactate that can be transferred to retinal neurons for metabolic support [88]. Their Oxphos capacity, however, is limited. It is suggested that this metabolic heterogeneity of the retina likely plays an important role in the cell-specific vulnerability to diabetes.

### 4.3. Substrate Selection and Energy Production

Tissues with a high metabolic rate, such as the heart, often utilize multiple energy sources (glucose, amino acids, fatty acids), which confers some degree of metabolic flexibility during periods of scarcity and surplus [91]. The retina also uses multiple fuel sources to generate the electron carriers NADH and FADH_2_, which donate their electrons directly to complex I and coenzyme Q-complex III, respectively, and ultimately establish the electrochemical gradient that drives ATP synthesis [81].

In addition to glucose, palmitate (C16:0), one of the most abundant fatty acids (FAs) in the human body, can also be used as a fuel substrate for retinal mitochondrial energy production [92]. In the retina, cellular FA uptake is mediated by the very low density lipoprotein receptor (VLDLR) that is expressed on both photoreceptor and RPE cells. Moreover, the expression of proteins involved in FA β-oxidation has been reported throughout the retina, including RGCs, photoreceptors, and Müller cells, albeit in varying amounts [93,94], indicating that retinal cells possess the machinery to oxidize lipids as fuel sources. FA oxidation is essential for retinal metabolism and function. In support of this concept, genetic mutations in specific enzymes involved in β-oxidation cause mitochondrial dysfunction, pigmentary retinopathy, and ultimately vision loss [95]. In mice, knockout of the peroxisome proliferative-activated receptor-α (PPARα), a nuclear receptor that modulates lipoprotein lipase expression and triglyceride metabolism, causes decreased lipid metabolism and retinal neurodegeneration [96]. Despite recognizing FA as a fuel source, the involvement of potential changes in FA β-oxidation remain largely unexplored in the retina in diabetes, a disease associated with increased FA availability. Nevertheless, two large-scale clinical trials known as the Fenofibrate Intervention and Event Lowering in Diabetes (FIELD) and Action to Control Cardiovascular Risk in Diabetes (ACCORD) studies have shown that the PPARα-agonist fenofibrate slows the progression of DR [97,98]. These findings raise the possibility that diabetes-induced alterations in mitochondrial FA β-oxidation may contribute to retinal dysfunction.

Further research has investigated the role of FA β-oxidation in the retinal microvasculature. Surprisingly, impaired FA oxidation in retinal endothelial cells neither results in energy depletion nor does it disturb redox homeostasis [99], suggesting that FA oxidation in these cells likely supports cellular functions in addition to ATP production. Using isotope labeling experiments, Schoors et al. [99] demonstrated that FA-derived carbon units are incorporated into aspartate (a nucleotide precursor) and eventually DNA. The same group also showed that blockade of carnitine palmitoyl transferase 1 (CPT1), the rate-limiting enzyme in FA β-oxidation, inhibits pathological neovascularization in mice. These data suggest a novel role of FA oxidation in endothelial cell proliferation and maintenance of the neurovascular unit. These findings raise the important question of whether, in diabetes, the retina exhibits a “metabolic switch” towards increased FA oxidation, similar to other high-energy consuming organs (e.g., heart, kidney). While such data are scarce, at least one study has indicated that the retina can increase the expression of FA oxidation enzymes in diabetic rats [100]. Subsequent studies will be needed, however, to characterize and quantify the relative contribution of individual energetic substrates to energy production in the diabetic retina.

Diabetes is characterized by decreased insulin action (a decrease in either secretion or sensitivity) and increased availability of energetic substrates (i.e., glucose and FAs). In the heart, also a highly energy consuming organ, increased availability of FAs is associated with a rapid decrease in glucose oxidation [101], thus reducing metabolic flexibility and increasing reliance on FAs for ATP production. In hepatocytes and adipocytes, increased availability of glucose leads to increased flux through the TCA cycle and increased production malonyl coenzyme A (malonyl-CoA), which inhibits CPT1 and spares lipids from β-oxidation. Similar to the heart, these changes indicate a decrease in metabolic flexibility. In the retina, glucose uptake occurs via both insulin-independent glucose transporter 1 (GLUT1) and insulin-dependent glucose transporter 4 (GLUT4) [102,103]. The retina also expresses a lipid sensor known as free fatty acid receptor 1 (Ffar1) (Figure 3) [92]. Ffar1 regulates insulin secretion in the islets of Langerhans [104] and neuronal function in the brain [105]. Interestingly, Ffar1 has been shown to downregulate GLUT1 expression in *VLDLR*-deficient photoreceptors, resulting in a dual glucose and lipid substrate uptake [92], which predictably leads to low levels of TCA cycle intermediates. Several TCA cycle intermediates including α-ketoglutarate and succinate have been shown to modulate the stabilization of hypoxia-inducible factor 1α [106]. In *VLDLR*-deficient photoreceptors, low α-ketoglutarate stabilizes hypoxia-inducible factor 1α and promotes neovascularization [92]. While similar studies have yet to be performed in the context of DR, these findings suggest multiple mechanisms by which the selection of energetic substrates, all plentiful in diabetes, may change retinal disease progression.

## 5. Alterations in Mitochondrial Oxidative Metabolism in the Neural Retina

### 5.1. Localization of Mitochondria in the Retina

An increasingly popular view is that diabetic milieu leads to the uncoupling of the retinal neurovascular unit [6], a concept that implies impaired crosstalk between neurons, glia, and the microvascular network [23]. While Müller cells rely primarily on glycolysis for ATP, those located in highly vascularized portions of the retina are rich in mitochondria [107], raising the possibility that mitochondrial dysfunction in Müller cells may have neurovascular consequences. Mitochondria also concentrate in the photoreceptor IS and the most external ends of the RPE (Figure 1), which likely reflects their migration toward the oxygen-rich choriocapillaris during neurodevelopment [108]. The specific distribution of mitochondria in the retina may explain, at least in part, the regional susceptibility to neurodegeneration and microvascular lesions in DR.

### 5.2. Diabetes Alters Mitochondrial Function in the Retina

The function of healthy and diseased mitochondria can be evaluated by measuring oxygen consumption rates in isolated mitochondria, retinal explants, or cultured retinal cells. However, due to limitations imposed by the small sample volume, many studies have resorted to studying mitochondrial function in retinal homogenate rather than individual cells or specific retinal layers. Retinal mitochondria exhibit a biphasic response to diabetes, characterized by an early and transient activation followed by a later decline. Masser et al. [100] showed that basal and ATP-linked oxygen consumption rates were significantly elevated in the retina of 3-month-old diabetic rats. In the same study, proteomic analysis revealed elevated levels of several FA β-oxidation enzymes and antioxidant proteins, suggesting a positive adaptive response of the retina to the diabetic milieu. This response mirrors the diabetic heart in that it indicates an energetic shift toward reliance on FA β-oxidation and metabolic inflexibility [101]. Another study reported increased mitochondrial oxygen consumption in diabetic rats at 3 weeks of hyperglycemia [109], which was associated with increased specific activities of complexes I, II, and III. Despite an increase in oxygen consumption and ETC complex activity, ATP generation was unchanged due to mitochondrial uncoupling, suggesting mitochondrial activation and inefficiency rather than mitochondrial defects. Similarly, in a model of spontaneous T2D in the cone-rich diurnal Nile rat, a short-term (2 month) hyperglycemia increased complex I-dependent mitochondrial respiration and was associated with increased cytochrome c access to cytochrome c oxidase, suggesting a change in composition or organization of the mitochondrial inner membrane [110]. Increased mitochondrial membrane permeability was confirmed in isolated mitochondria from retinas of Zucker diabetic fatty rats with 6 weeks of persistent hyperglycemia and was associated with a concomitant decrease in mitochondrial complex III-specific activity [111].

Osorio-Paz et al. [109] reported that longer (approximately 6 weeks) durations of STZ-induced diabetes in rats caused a decreased cytochrome c-reducing activity of complex III, while complexes II and IV were hyperactive when measured in isolated retinal mitochondria. These changes in ETC complex-specific activities reflect a decrease in oxygen consumption of retinal mitochondria energized with a combination of energetic substrates (glutamate and malate) that are generating NADH to be oxidized by complex I. Intriguingly, the generated ATP was unchanged. Mitochondrial respiration in the presence of substrates feeding electrons into complex I (NADH pathway) and II (succinate pathway) involves complexes III and IV, as well as mobile electron carriers such as co-enzyme Q and cytochrome c. Therefore, it is expected that a change in individual components will affect the whole Oxphos pathway. However, the effect of individual components on the whole integrative function depends upon the control of that component on the Oxphos. In comparison with heart, where the impact of individual components on the Oxphos and ATP synthesis has been determined, this control has yet to be investigated in the normal and diabetic retina. While it is unclear if complex III defect is limiting for the Oxphos in the diabetic retina, a decreased complex III activity led to increased superoxide in a mouse model of STZ-induced diabetes; both were normalized by overexpressing the mitochondrial antioxidant enzyme, manganese superoxide dismutase (MnSOD) [112].

In contrast with short-term diabetes, a long-term (18 months) sustained hyperglycemia in T2D Nil diurnal rats caused a decrease in NADH-supported mitochondrial respiration accompanied by an increase in succinate contribution to the maximal Oxphos capacity in the whole retinas, thus confirming a partial decline in mitochondrial bioenergetics [110]. As NADH-induced mitochondrial respiration is supported by complex I, co-enzyme Q, complex III, cytochrome c, and complexes IV and V, these results suggest a potential defect in any of these Oxphos subunits.

Mitochondrial DNA (mtDNA) follows a similar response in the diabetic retina. Alterations in the morphology and function in the neural retina occurring within the first 3 months of diabetes in rats are not associated with changes in mtDNA in isolated retinal synaptosomes. These findings suggest that alterations of mtDNA in synaptosomes are not causative for the early neural retina dysfunction in diabetes [100]. In addition, in a model of T1D in rats, while an elevated oxidative stress was detected as early as 15 days of diabetes, mtDNA damage was observed much later at 6 months due to inactivation of the DNA repair/replication enzymes [113]. A similar temporal relationship was observed in endothelial cells exposed to high glucose [114]. These data suggest that the oxidative damage of mtDNA is fully compensated in early stages of diabetes while the altered mtDNA in later stages leads to a decline in mitochondrial transcription and secondary ETC defects. A summary of the mitochondrial alterations in DR is presented in Table 1 and Table 2.

## 6. Mitochondria-Derived Oxidative Stress in the Diabetic Retina

Oxidative stress is a critical component of altered homeostasis across multiple cell types involved in DR [141]. ROS are defined by the presence of a highly reactive oxygen molecule, and are generated as normal byproducts of cellular redox reactions. Mitochondria generate superoxide during oxidation-reduction reactions as some electrons may leak to univalently reduce molecular oxygen. A number of mitochondrial antioxidant defense mechanisms are in place to prevent the increase in oxidative stress, including MnSOD, catalase, reduced glutathione, and thioredoxin. Mitochondria are both the producers and targets of oxidative stress, with the latter altering mtDNA and mitochondrial proteins leading to mitochondrial dysfunction. Defects in the ETC further amplify the risk of increased oxidative stress. In the retina, this self-perpetuating cycle has been referred to as the “metabolic memory” phenomenon, a hypothesis that is supported by the persistence of altered mtDNA, decreased activity of ETC complexes, and increased oxidative stress throughout the retina despite the reinstitution of good glycemic control [142].

The proposition that increased oxidative stress is a key pathogenic factor in the development of DR is supported by finding of insufficient antioxidant defenses in diabetic patients [143]. Antioxidant approaches alleviate diabetes-induced vascular lesions in the retina [112,144,145]. Moreover, the causal link between increased mitochondrial-generated oxidative stress and retinal microvascular disease in diabetes is supported by the effect of overexpressing the mitochondrial antioxidant enzyme, MnSOD, to decrease the number of acellular capillaries in diabetic mice [112]. Therefore, mitochondria are directly implicated in the development of diabetic microvascular lesions.

Mitochondrial dysfunction in retinal endothelial cells has been identified as the upstream contributor to diabetic vascular disease in diabetes [146,147]. Recent evidence indicates that the neurovascular unit is functionally affected before the onset of retinal microvascular disease, and that the neural retina is affected by oxidative stress originating from the photoreceptors [148]. The role of photoreceptors as oxidative stress generators is supported by the observation that human patients with photoreceptor degeneration and retinitis pigmentosa have less severe DR than diabetics with intact photoreceptors as well as diabetic mice lacking photoreceptors due to opsin deficiency [34]. The work of Du et al. [34] unequivocally identified photoreceptors as the major source of superoxide generated by retinas of diabetic mice, and showed that mitochondria contributes to at least 50% of oxidative stress, thus complementing the NADPH oxidase. Their deletion inhibited the expected increase in superoxide and inflammatory proteins in the remaining retina in diabetic mice. Of note, Müller cells cultured in high glucose also exhibit increased oxidative stress [149], but their contribution to DR in vivo is unknown.

There are two potential mechanisms explaining the increased superoxide production by the diabetic photoreceptors. The first hypothesis is that defects of the mitochondrial ETC interrupt the normal electron flow to fully reduce oxygen, thus leading to accumulation of electrons at sites within the ETC, which are accepted by oxygen to generate superoxide [150]. The ETC sites prone to leak electrons to oxygen are complexes I and III [151,152]. In support of this hypothesis, we recently reported that correcting the electron flow within the complex I-deficient ETC decreased oxidative stress and photoreceptor damage [153]. The second hypothesis is that an early increase in mitochondrial oxidative phosphorylation fed by an increased FA β-oxidation brings additional sites of electron leak to oxygen, as was shown for the heart [154,155] and kidney tubules in diabetes [156]. These possibilities are not mutually exclusive, but they are yet to be investigated in the retina.

Superoxide is dismutated to hydrogen peroxide, a highly permeable compound that may support the crosstalk between retinal cells and affect neighbor cells, thus providing a link between neural retina and microvasculature. However, this crosstalk has not yet been investigated in the retina. Oxidative stress increases the expression of pro-inflammatory proteins [157] and enhance retinal inflammation [158] that contributes to early DR [159]. In addition, in the microvasculature, increased oxidative stress is also associated with increased apoptosis [114,160,161], further compromising the integrity of the neurovascular unit.

## 7. NAD Pool and the NADH/NAD^+^ Redox Ratio in the Diabetic Retina

### 7.1. NAD Pool

Nicotinamide adenine dinucleotide (NAD) is a coenzyme for redox enzymes, shuttling electrons from glycolysis and the TCA cycle to complex I in the ETC. The oxidized form, NAD^+^, is also a co-substrate for non-redox reactions such as those catalyzed by sirtuin (SIRT) and poly (ADP-ribose) polymerase (PARP) families of proteins [162,163]. This link represents a highly conserved mechanism by which redox status influences a wide range of cellular and metabolic functions, including cell signaling, DNA transcription, and programmed cell death. Recent work by Lin et al. [164] suggest that NAD is essential for vision. In this study, specific deficiency of NAD in rod photoreceptor for 6 weeks led to massive atrophy of the entire neurosensory retina, affecting the microvasculature, RPE, and optic nerve. The complete absence of the outer nuclear layer (photoreceptor nuclei) indicates that cone photoreceptors are also secondarily affected by rod NAD deficiency. These results strongly suggest that retinal photoreceptors are essential for the integrity of the whole retina. Although similar studies have not yet been conducted in the context of DR, these findings suggest that alterations of the photoreceptors precede the vascular regression in diabetes. During a short (3-week) period of NAD deficiency, mitochondrial morphology was maintained as normal, whereas at 4 weeks, mitochondrial cristae were lost, and photoreceptor OS were disrupted. Metabolomic analysis showed that NAD deficiency causes dysregulation of multiple metabolic pathways including the TCA cycle, mitochondrial protein biosynthesis, and propionate metabolism with accumulation of acylcarnitines, and also decreased ATP production. Both glycolysis and mitochondrial Oxphos were affected. These results highlight the critical role of metabolism and bioenergetics to maintain the photoreceptor integrity. The same research group identified the decreased retinal NAD pool as an early feature in retinal disease caused by STZ-induced diabetes at 3 weeks of sustained hyperglycemia [164]. NAD deficiency caused photoreceptor death and diminished rod ERG recordings. These data support the concept that mitochondrial dysfunction in photoreceptors in the neural retina proceed the vascular regression in the diabetic retina.

### 7.2. NADH/NAD^+^ Redox Ratio

The “hyperglycemic pseudohypoxia” hypothesis [165,166] suggests that diabetes is associated with an increased cellular NADH/NAD^+^ redox ratio attributed to an increased flux through the polyol pathway and resulting in altered metabolism and neurovascular dysfunction. The polyol pathway is a two-step reaction involving the reduction of glucose to sorbitol and the subsequent oxidation of sorbitol to fructose [167]. The rate-limiting step in this pathway is catalyzed by aldose reductase, which is expressed in all cells and utilizes NADPH as an electron donor. Importantly, aldose reductase is activated by hyperglycemia. The second step in the polyol pathway is catalyzed by sorbitol dehydrogenase, which uses NAD^+^ as an oxidizing agent to produce fructose and thus increases the NADH/NAD^+^ redox ratio. Studies have shown that both sorbitol and fructose accumulate in diabetic tissues, including the retina [168,169], suggesting that the polyol pathway may contribute to oxidative stress in DR. This finding is supported by the observation that genetic knockout of aldose reductase protects retinal endothelial cells from oxidative stress [170].

The redox state is compartmentalized between cellular organelles. The cytosolic NADPH/NADP^+^ is maintained in a reduced state necessary for drive biosynthetic and antioxidant processes. In energized mitochondria NADH exceeds NAD^+^ to provide electrons for the ETC while the cytosol has a higher NAD^+^, reflecting a relatively oxidized redox state [171]. Mitochondrial NADH/NAD^+^ redox ratio is closely related with mitochondrial function. We recently reported that a mitochondrial complex I defect directly results in an increased NADH/NAD^+^ ratio in cultured photoreceptor cells [153]. Diederen et al. [172] found no significant differences in NADH/NAD^+^ redox ratios in the whole retina of 6-month-old STZ-induced diabetic mice. The redox status in specific cellular organelles including mitochondria was not assessed in this study. It may be predicted that mitochondrial redox state is unchanged in early stages of DR when considering the biphasic response of retinal mitochondria to the diabetic milieu. An early enhanced mitochondrial function would be expected to maintain a normal NADH/NAD^+^ redox ratios until ETC defects begin to manifest. While complex I and IV defects are associated with the accumulation of NADH and an increased NADH/NAD^+^ ratio [173,174], measures taken to correct ETC defects can be used to decrease NADH and restore redox balance [175]. These data are consistent with the hypothesis that ETC defects cause an increase in NADH and a reduced redox microenvironment in retinal mitochondria.

A potential mechanism that may increase cellular NADH/NAD^+^ redox ratio is the activation of poly (ADP-ribose) polymerases (PARPs), a family of proteins best known for their role in DNA repair. PARPs are activated in response to DNA damage and catalyze the transfer and polymerization of ADP-ribose to DNA repair enzymes. This reaction requires NAD^+^, leading some to hypothesize that PARP activation could lead to NAD^+^ depletion and altered redox homeostasis. *PARP*-deficient mice are protected against diabetes [176], and exhibit preserved redox homeostasis and mitochondrial function [177]. The mitochondrial protective effect is mediated by activating the NAD^+^-dependent deacetylases, sirtuins (SIRTs). Among the large family of SIRT proteins, SIRT1 is an extramitochondrial protein with a wide range of functions in both metabolism and aging. Recent work by Mishra et al. [178] have shown that the *SIRT1* promoter is hypermethylated in STZ-induced diabetic mice. *SIRT1* overexpression protected the mice against mitochondrial damage, neural dysfunction, RGC degeneration, and blood–retinal barrier breakdown. A role of mitochondrial sirtuins in retinal disease is supported by the finding that genetic knockout of *SIRT3*, a mitochondrial SIRT, mirrors NAD^+^ deficiency and leads to early and rapid retinal degeneration [164].

The immediate consequence of an increased mitochondrial NADH/NAD^+^ redox ratio is reductive stress (increased NADH) and NAD^+^ deficiency, which is detrimental to photoreceptor integrity [153]. Mitochondrial production of ROS is largely governed by the NADH/NAD^+^ ratio, as an increased [NADH] slows the ETC flux. The antioxidant defense is supported by the NADPH/NADP^+^ redox ratio. NADPH is a potent reducing agent involved the regeneration of antioxidant compounds such as reduced glutathione. Importantly, the mitochondrial NADH/NAD^+^ and NADPH/NADP^+^ redox couples are linked by nicotinamide nucleotide transhydrogenase (NNT), an enzyme that leverages the proton-motive force in the oxidation of NADH and the simultaneous reduction of NADP^+^ (Figure 4A). NNT maintains a NADPH/NADP^+^ ratio several-fold higher than the NADH/NAD^+^ ratio, and thus is a physiologically relevant source of NADPH that drives the reduction of H_2_O_2_ [179]. While NNT is reported to be expressed exclusively in cardiac tissue [180], we provide evidence here that the NNT protein is also expressed in the retina (Figure 4B). The role of NNT to maintain the mitochondrial redox state and antioxidant defense is yet to be determined.

## 8. Alterations in Mitochondrial Turnover

### 8.1. Mitochondrial Biogenesis and Mitophagy

The balance between mitochondrial formation and destruction regulates the cellular mitochondrial mass. Mitochondrial biogenesis is the cellular process to increase total mitochondrial content. This process relies on the coordinated action of cell signaling molecules, molecular chaperones, and transcription factors, all of which working in tandem to replicate the mitochondrial genome and proteome. One of the most upstream factors is the peroxisome proliferator-activated receptor gamma coactivator 1-α (PGC-1α), which has been referred to as the “master regulator” of mitochondrial biogenesis [183]. Among its many downstream targets is mitochondrial transcription factor A (TFAM), which is translocated to the mitochondrial matrix and initiates mitochondrial genome replication. In the past decade, a series of studies conducted by Santos et al. [127,129,184] have shown that mitochondrial biogenesis is altered in both experimental and human DR. Specifically, Santos et al. [129] found that nuclear-mitochondrial translocation of TFAM is impaired within 12 months of STZ-induced diabetes. Subsequent experiments by the same group determined that TFAM is ubiquitinated and targeted for proteasomal degradation, and its translocation to the matrix is impaired [128]. Overexpression of MnSOD or administration of the exogenous antioxidant lipoid acid had a positive impact on mitochondrial localization of TFAM and mtDNA copy number [127,129], further supporting a role of oxidative stress as an upstream regulator of mitochondrial biogenesis. An important limitation, however, is that most of these studies were performed in retinal homogenate or endothelial cells. Thus, whether these specific findings translate to specific cell populations within the neural retina is an important question that remains to be explored.

Mitophagy is a specialized form of macro-autophagy by which damaged or excessive mitochondria are selectively targeted for lysosomal degradation. Several groups have reported that Müller cells grown in high glucose exhibit enhanced mitophagy [119,120,121,124]. This phenomenon is thought to occur in part as a consequence of hyperglycemia-induced expression of thioredoxin-interacting protein (TXNIP), which binds to and inhibits the antioxidants thioredoxin 1 and thioredoxin 2. Hyperglycemia-induced expression of TXNIP is observed in the vasculature, pericytes, and the RPE [130,185], suggesting a conserved mechanism across cell types. Knockdown of TXNIP reduces oxidative stress, improves ATP synthesis, and restores mitophagic flux [119]. Some of these findings have since been validated in the db/db mouse model by Zhou et al. [124]. When considered together with the work of Santos et al. [114,127,128,129,186], these findings suggest that DR is characterized by a gradual decrease in mitochondrial content, both due to impaired biogenesis and enhanced mitophagy. Recent work by Hombrebueno et al. [121] suggests that these processes have a temporal relationship. Using the spontaneous *Ins2^Akita^* diabetic mouse model, Hombrebueno et al. [121] observed enhanced PTEN-induced kinase (PINK1)-dependent mitophagy in both Müller cells and photoreceptors within the first 2 months of diabetes. While increased mitochondrial biogenesis can compensate for enhanced mitophagy, compensatory mechanisms begin to fail at 8 months of diabetes, resulting in decreased mitochondrial mass.

### 8.2. Fusion–Fission Dynamics in the Retina

Mitochondria exist in a constant flux of fusion and fission, which is necessary to respond to the energy requirements of the cell (for a review, see [187]). This process is regulated by mitofusin-2 (Mfn2) and dynamin-related protein 1 (Drp1), two antagonistic GTPases that regulate fusion and fission, respectively. In human DR, Mfn2 protein levels are reduced, while Drp1 is increased, suggesting an imbalance between these two GTPases that favors mitochondrial fission, reduced mtDNA, and possibly decreased ATP synthesis [188]. These findings have also been observed in Müller cells and photoreceptors grown in high-glucose conditions [119]. Although the exact mechanism is largely unexplored, a shift toward mitochondrial fission may also be due to oxidative damage. In support of this hypothesis, administration of melatonin, which is well-known for its antioxidant properties, preserved mitochondrial fusion in photoreceptors both in vitro and in vivo [118]. Alternatively, enhanced mitochondrial fission may be due to increased methylation at the *MFN2* promoter site, as shown in endothelial cells [136]. However, this finding alone does not explain the reported increase in Drp1, which independently favors mitochondrial fission. This area of research is still in its preliminary stages, and subsequent studies will be necessary to confirm its significance in DR.

## 9. Consequences of Increased Oxidative Stress in the Neural Retina

### 9.1. Diabetic Milieu Alters Ion Channel Homeostasis in the Retina

Early in its course, diabetes causes a paradoxical closure of the L-type calcium ion channels (LTCCs) in the dark, as suggested by manganese-enhanced magnetic resonance imaging (MEMRI) studies that have shown that photoreceptor uptake of manganese (a calcium surrogate) is significantly reduced in the dark-adapted rodents [189]. Because these ion channels are essential for the regulated release of neurotransmitters at the photoreceptor synapses, paradoxically closed photoreceptor LTCCs in the dark have significant functional consequences. Alterations in ion channel homeostasis have also been reported at the level of the mitochondria. The expression of the mitochondrial calcium uniporter (MCU), which plays important roles in calcium buffering and ion channel homeostasis, is decreased in photoreceptors grown in high glucose conditions [190]. Retinal neurons cultured in high glucose exhibit increased mitochondrial calcium load, associated with depolarization of mitochondrial membrane and ROS generation. Similar observations were made in retinas from 9-week-old diabetic rats [191].

Long-term diabetes causes mitochondrial ETC defects and decreases mitochondrial respiration efficiency in the retina, with both being canonical causes of energy deficit. However, available data do not support the hypothesis that ATP deficit is responsible for the dysfunction in ion channels as retinal [ATP] is unchanged in diabetes [192]. Decreasing oxidative stress in diabetic rodents with copper/zinc superoxide dismutase (Cu/Zn SOD) overexpression or lipoic acid administration corrected the diabetes-induced ion flux abnormalities in photoreceptors in the dark [145,193], suggesting that abnormalities in ion homeostasis are induced by increased oxidative stress. Studies showing that diabetes elevates oxidative stress in the retina by both promoting ROS production and suppressing the antioxidant defense [117] also provide strong support for this hypothesis.

### 9.2. Apoptosis

The intrinsic (mitochondrial) pathway of apoptosis is initiated by increased permeabilization of the mitochondrial outer membrane and activation of the apoptotic signaling cascade. Notably, the intrinsic pathway is induced by increased oxidative stress [194], making this pathway highly relevant in DR, as virtually all cell types in the retina experience hyperglycemia-induced oxidative stress [116,123,164,195]. RGCs [133] and the retinal microvasculature in particular have been shown to undergo oxidative stress-induced apoptosis [114,160]. Several groups have reported that administration of exogenous antioxidants preserved mitochondrial integrity and prevent cell death in the diabetic retina [116,133]. These findings reiterate the concept that oxidative stress is an early event in the pathogenesis of DR, whereas cell death generally occurs as a later and secondary event.

## 10. Therapeutic Implications

### 10.1. Therapies Focused on Maintaining the Integrity of Retinal Mitochondria

The Diabetes Control and Complications Trial showed that intensive insulin replacement therapy reduces the incidence and slows the progression of DR [5]. However, the incidence of DR remains high, and many patients still progress to PDR despite advances in diabetes care. Accordingly, second-line treatments are frequently necessary to manage the later complications of DR. These therapies are highly effective in managing the sight-threatening complications of DR. Thus, it is critical to better understand the early stages of DR and develop new therapies to prevent its progression. The need for new therapies in early stages of DR is further highlighted by the “metabolic memory” phenomenon (for a review, see [2]), which may be a consequence of persistent mitochondrial damage and oxidative stress [142]. This hypothesis is consistent with the benefit of mitochondrial targeted therapy to preserve mitochondrial integrity and vision in experimental DR.

An important antioxidant therapy is the Szeto–Schiller (SS) tetra-peptide, SS-31 (elamipretide), which is concentrated in the inner mitochondrial membrane and selectively stabilizes cardiolipin [196], a phospholipid critical for mitochondrial integrity and function, which is prone to oxidative damage in diabetes [197]. SS-31 enhances the interaction between cytochrome c and cardiolipin to facilitate better electron transfer from complex III to complex IV, and reduces mitochondrial oxidative stress [198,199]. Evidence from animal studies suggests that SS-31 could reduce the risk of vascular disease in diabetes, as administration of SS-31 alleviated the microvascular retinal disease in rodent models of DR [200,201], and increased SIRT1 while ameliorating retinal inflammation in rodent and human subjects with T2D [202]. However, human studies have shown limited efficacy to date, and no studies have been performed in patients with DR.

Numerous studies have suggested that oxidative stress may be modulated by uncoupling proteins (UCPs), a family of proteins named for their ability to uncouple electron transport from ATP synthesis. These proteins decrease mitochondrial ROS production by decreasing the electrochemical gradient [203], a process called “mild uncoupling”. The theory is based on the observation of Korshunov et al. [204] that ROS generation increases in an exponential manner when mitochondrial membrane potential exceeds a threshold that is higher than that corresponding to the mitochondrial energetic state in in vivo settings. The hypothesis is also based on the assumption that the diabetic milieu increases the availability of energetic substrates (glucose, FAs) to the retina, which are oxidized to increase mitochondrial membrane potential (“hyperpolarization”). This hypothesis is supported by the observation that *UCP2*-deficient mice exhibit increased ROS generation [205], whereas overexpression of the *UCP2* gene preserves mitochondrial function in human umbilical endothelial cells [206]. In the retina, UCP2 protein levels and activity are increased in retinal endothelial cells grown in high-glucose conditions [207], which may indicate a compensatory response to oxidative stress. However, administration of the uncoupling agent niclosamide ethanolamine has shown no benefit in the treatment of diabetes or its complications in db/db mice [208] indicating the lack of benefit of uncouplers in in vivo settings. A potential explanation is that Oxphos is a process regulated by energy demand rather than substrate availability. In diabetes, increased substrate availability does not increase oxidative metabolism that exceeds ATP synthesis [109], a mechanism that would lead to “hyperpolarization”. While UCPs may be an important endogenous defense mechanism in the setting of oxidative stress, their therapeutic utility is unclear.

Improving the efficiency of electron flow within the defective mitochondrial ETC may be an optimal approach to relieve the increased electron density at specific ETC sites and eliminate the risk of oxygen univalent reduction and superoxide formation. Methylene blue, a redox compound that provides an alternative electron route between mitochondrial complex I and cytochrome c [209], preserves mitochondrial and photoreceptor integrity by preventing oxidative stress in a model of complex I defect [153] and experimental diabetes [34].

Idebenone, a synthetic benzoquinone that mediates electron transfer to complex III by bypassing complex I [210], is a free radical scavenger [211], reduces intracellular ROS and increases ATP production in complex I-defective cells [212,213,214], and promotes an increase in mitochondrial mass by regulating mitophagy [215]. Its mitochondrial protective effects have recently been investigated as a drug therapy for Leber’s hereditary optic neuropathy, a rare genetic mitochondrial disease that causes rapid and progressive bilateral vision loss in young adults. A 24-week multi-center double-blind, randomized, placebo-controlled trial in patients with Leber’s hereditary optic neuropathy show a mild benefit in visual acuity [216] that was confirmed when treatment started 5 years after the diagnosis [217]. The benefic outcomes are considered a result of idebenone to restore the bioenergetics in the remaining dysfunctional RGC. Evidence for efficacy of idebenone in human patients with primary or acquired mitochondrial defects are still limited, and restrict its use in DR.

### 10.2. Clinical Trials and the Role of Antioxidants in the Management of DR

Despite the evidence that antioxidants can slow the progression of DR, clinical trials have had mixed results. This topic was very recently reviewed by Garcia-Medina et al. [218], but is summarized here for completeness. The most successful interventions have been those that use combined antioxidant therapy (CAT). The Diabetes Visual Function Supplement Study showed that CAT may improve visual acuity and contrast sensitivity among participants with T1D and T2D without clinically detectable retinopathy or with mild non-proliferative diabetic retinopathy (NPDR) [219]. This finding is supported by the prior work of Hu et al. [220], who showed that patients with NPDR have lower levels of lutein and zeaxanthin, and further demonstrated that supplementation with these antioxidants reduces oxidative stress and may improve visual function. Although the previous two studies suggest a role of CAT in the management of DR, the conclusions are limited by the short study duration (6 and 3 months, respectively) relative to the chronicity of DR. To date, the longest trial that has been conducted was a 5-year follow-up of patients taking a commercially available multi-vitamin showing that antioxidants may slow the progression of DR as detected by ophthalmic examination [221]. However, in contrast to the above studies, the investigators did not observe a significant improvement in visual acuity. These discrepancies highlight the need for additional studies to find better therapeutic strategies that decrease the mitochondrial ROS generation by preserving mitochondrial function rather than scavenging the already generated ROS.

## 11. Conclusions

The pathogenesis of DR is complex and likely involves the simultaneous dysregulation of multiple metabolic and signaling pathways throughout the retinal neurovascular unit. Alterations in mitochondrial function has broader consequences than changes in ATP content. Increased oxidative stress and alterations in the redox balance are interrelated mechanisms that are altered by diabetes, and their effect on retinal structure and function in diabetes is yet to be explored. The benefit of maintaining retinal energetic flexibility and optimal fuel selection between plentiful competing substrates (glucose versus FA) in diabetes remains largely unexplored, and may represent a promising area of research. Although the present review focuses on the roles of mitochondrial dysfunction and oxidative stress, cellular dysfunction in the retina can take many forms, including neuroinflammation and blood–retinal barrier breakdown. These processes likely occur in parallel and thus future studies should adopt a comprehensive approach that appreciates the interconnectedness of the retina.

## Figures and Tables

**Figure 1 antioxidants-09-00905-f001:**
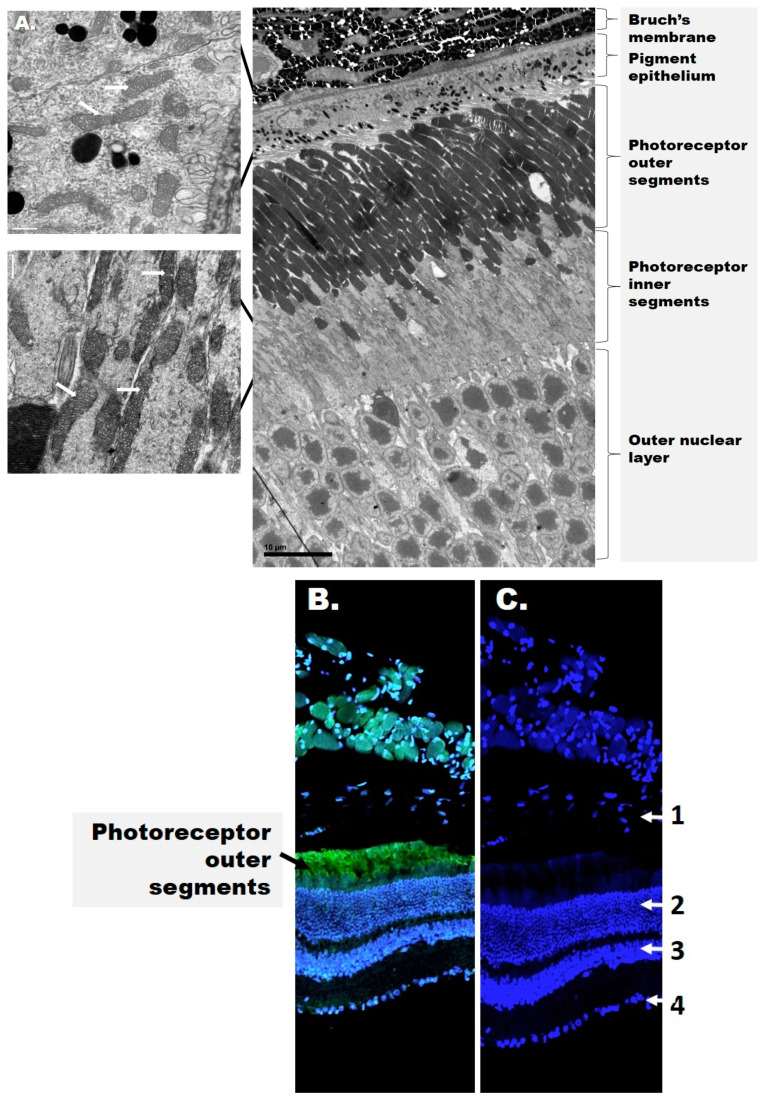
The structure of the retina. (**A**) Electron microscopy images of the mouse outer retina. Mitochondria are shown in the figure inset and are indicated by white arrows. (**B**) Confocal image of the mouse retina depicting rhodopsin (green) and cell nuclei (blue). (**C**) Distribution of cell nuclei (blue) in the mouse retina. The numbers represent the retinal layers: 1—retinal pigment epithelium (RPE, detached); 2—outer nuclear layer; 3—inner nuclear layer; 4—ganglion cell layer. Rhodopsin (green fluorescence) is present in stacks of membranous disks of the photoreceptor outer segments (OS). 4′,6-diamidino-2-phenylindole (DAPI, blue) stains the nuclei in all nuclear layers and the RPE.

**Figure 2 antioxidants-09-00905-f002:**
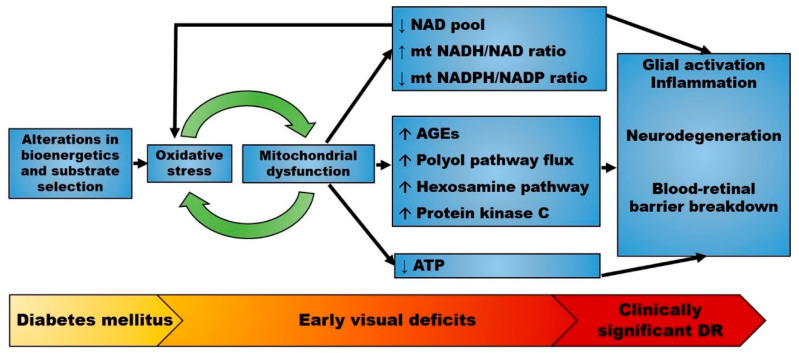
Proposed pathophysiology of diabetic retinopathy (DR). The early stages of diabetes mellitus are characterized by alterations in bioenergetics and substrate selection in a variety of cell types. In the retina, these changes cause oxidative stress and are associated with early visual deficits such as impaired contrast sensitivity. Mitochondrial oxidative stress alters mitochondrial metabolism and upregulates multiple seemingly independent pathways leading to retinal disease. Mitochondrial dysfunction also changes the redox state that further enhances oxidative stress. Therefore, mitochondrial-generated oxidative stress may precede overt neurodegeneration and microvascular disease. Abbreviations: AGEs, advanced glycation end products; DR, diabetic retinopathy; mt, mitochondrial. ↑: increased; ↓: decreased.

**Figure 3 antioxidants-09-00905-f003:**
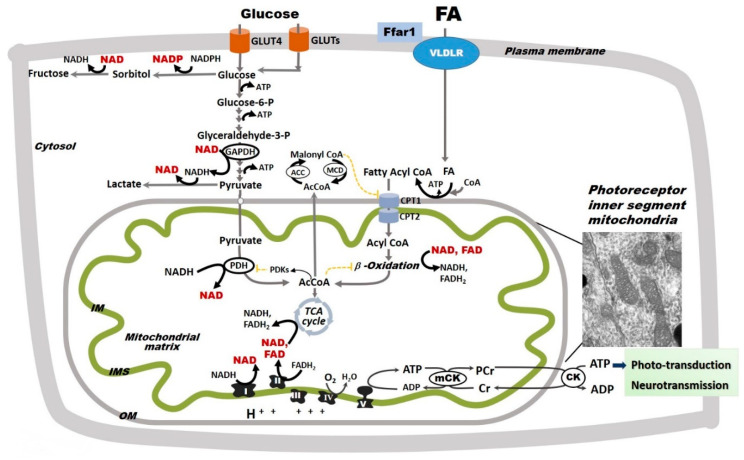
Glycolysis and oxidative phosphorylation in the retina. The retina relies on glycolysis and mitochondrial oxidative phosphorylation (Oxphos) as sources of ATP. Glucose uptake into retinal cells occurs via insulin-dependent glucose transporter 4 (GLUT4) and insulin-independent glucose transporter 1 (GLUT1). Fatty acid (FA) uptake is not hormonally regulated, but rather potentially driven by circulating availability (FA_circ_) [89,90]. In the retina, FA uptake is regulated by a lipid sensor, the free fatty acid lipid receptor 1 (Ffar1), and mediated by the very low density lipoprotein receptor (VLDLR). In retinal cells, glucose follows multiple metabolic pathways, including glycolysis and the polyol pathway, the latter of which leads to the production of sorbitol and ultimately fructose. Pyruvate is either converted to lactate or transported into mitochondria where it is converted by pyruvate dehydrogenase (PDH) to acetyl coenzyme A (AcCoA), which enters the tricarboxylic acid (TCA) cycle. PDH is inhibited by pyruvate dehydrogenase kinases that are activated by excessive acetyl-CoA and nicotinamide adenine dinucleotide (NADH). For simplicity, other glucose metabolic pathways are not shown. FAs, which are released from triglycerides and imported into retinal cells via the VLDLR, are converted to fatty acyl-CoA, shuttled into the mitochondria via carnitine palmitoyltransferases 1 and 2 (CPT1 and 2), and oxidized via FA β-oxidation. FA β-oxidation yields NADH, flavin adenine dinucleotide (FAHD_2_), and AcCoA, which are further oxidized by the electron transport chain (ETC) complexes in the process of Oxphos with ATP synthesis. FA β-oxidation is inhibited by malonyl-CoA (an intermediate of FA synthesis), FADH_2_/FAD^+^ ratios, and NADH/NAD^+^ ratios. Malonyl-CoA is degraded by malonyl-CoA decarboxylase (MCD), thus decreasing its inhibitory effect on CPT1. Although described in other organs, these regulatory steps are yet to be identified in the retina. Mitochondrial Oxphos provides the bulk of retinal ATP. As electrons transfer NADH and FADH_2_ to molecular oxygen by ETC complexes, an electrochemical gradient is built across the mitochondrial inner membrane (IM). This gradient is used by the complex V to produce ATP. Mitochondria-generated ATP is transferred to the cytosol by the creatine kinase (CK) shuttle to sustain the normal functions of the retinal cells. The inset included here is an electron micrograph of a mouse rod photoreceptor and shows the inner segment mitochondria. For simplicity, the nicotinamide nucleotide transhydrogenase, a mitochondrial inner membrane enzyme that reduces nicotinamide adenine dinucleotide phosphate (NADPH^+^) by oxidizing NADH and using the mitochondrial proton-motive force, is not shown in this figure. The oxidized NAD^+^ and NADP^+^ are shown in red.

**Figure 4 antioxidants-09-00905-f004:**
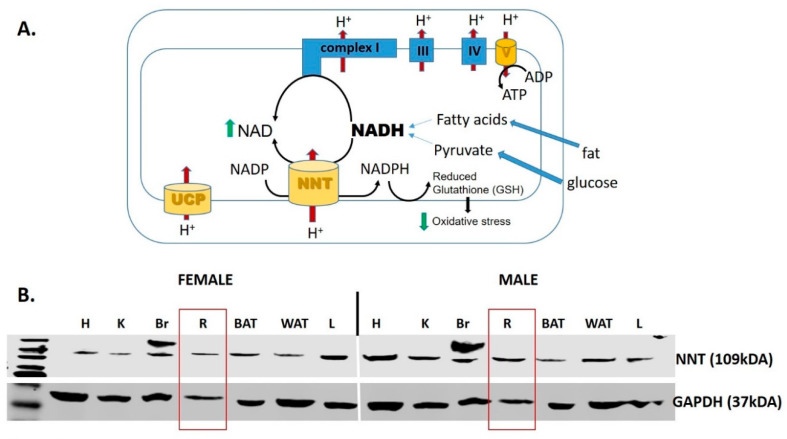
Nicotinamide nucleotide transhydrogenase (NNT). (**A**) NNT is a mitochondrial enzyme that oxidizes NADH, and therefore supplements complex I in the process of regenerating NAD^+^. In addition, the enzyme forms NADPH that is critical to maintain the antioxidant potency of mitochondria by maintaining the reduced glutathione (GSH) [181,182]. (**B**) Western blot analysis of the NNT protein expression in a variety of tissues in both male in female mice. Abbreviations: UCP, uncoupling protein; H, heart; K, kidney; Br, brain; R, retina; BAT, brown adipose tissue; WAT, white adipose tissue; L, liver.

**Table 1 antioxidants-09-00905-t001:** Summary of investigations into altered mitochondrial homeostasis in the diabetic retina by cell type *.

Cell Type	Oxidative Stress	Mitochondrial Morphology	Mitochondrial Turnover	References
RGCs	✔	✔	Unknown	[115,116,117]
Bipolar cells	✔	✔	Unknown	[117]
Müller cells	✔	✔	✔	[118,119,120,121,122,123,124]
Photoreceptors	✔	✔	✔	[121]
Endothelial cells	✔	✔	✔	[113,114,125,126,127,128,129,130,131]

* Not an exhaustive list. Abbreviations: RGC, retinal ganglion cell. ✔: this feature was confirmed in the respective cells, as supported by references.

**Table 2 antioxidants-09-00905-t002:** Summary of electron transport chain protein expression and activity in the diabetic retina by cell type *.

Cell Type	Complex I	Complex II ^†^	Complex III	Complex IV	Complex V (ATP Synthase)	References
RGCs	↓	↓	Unknown	↓	Unknown	[132,133]
Bipolar cells	Unknown	Unknown	Unknown	Unknown	Unknown	
Müller cells	Unknown	↔	Unknown	↔	Unknown	[119]
Photoreceptors	Unknown	Unknown	Unknown	Varies	Unknown	[121]
Retinal homogenate	Biphasic ^‡^	Biphasic	Biphasic	Biphasic	Varies	[100,109,110,111,112,121,134]
Pericytes	Unknown	↓	Unknown	Unknown	Unknown	[135]
Endothelial cells	↓	↓	↓	↓	↓	[129,136,137,138,139]

* Not an exhaustive list. ^†^ The activity of complex II was inferred in this study from the 3-(4,5-dimethylthiazol-2-yl)-2,5-diphenyltetrazolium bromide (MTT) assay, which measures the ability of succinate dehydrogenase in complex II to reduce tetrazolium salts [140]. ^‡^ Biphasic changes in electron transport complex activity are characterized by early activation followed by a later decline. The exact timing varies by model. Abbreviations: RGC, retinal ganglion cell. ↓: decreased; ↔: unchanged.

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
