# Peer review of "Diabetic Retinopathy: The Role of Mitochondria in the Neural Retina and Microvascular Disease"

_antioxidants, 2020, doi:10.3390/antiox9100905_

Round 1

Reviewer 1 Report

Dear Editor,

It is a lengthy review manuscript in which authors attempted to prove the concept that the real origin of problems encountered in DR starts in photoreceptors, notably by the massive production of ROS. Authors try to prove that vascular problems develop in later stages of the disease and they are preceded by huge mitochondrial abnormalities of outer retinal layers that start to occur in early stages of DR. Personally, I believe it make sense however, it is not absolutely sure that this concept presents the only one possible scenario in this complicated disease.

The manuscript is not so explicitly written, thus, it requires significant corrections before it can be published. Overall, the ms is quite messy.

What should be done is mainly to shift accents.

First part of the manuscript is far too long and although it presents physiology of the retina and pathophysiology of DR quite well, only late section 6 in page 12 is devoted to oxidative stress in the diabetic retina.

Just to remind you, authors promised in Abstract: " to discuss the recent developments related to alterations in mitochondrial substrate selection and bioenergetics............"  and " emerging therapeutic approaches and future research directions......." Especially this last part is underdeveloped.

It should be more "cutting edge" otherwise it adds very little to the current knowledge. I would avoid repeated ideas and sentences in which this ms is quite rich.

References is a misunderstanding, there are lacking authors, titles, abbreviations of the names of journals, volumes etc. It is very messy.

MAC and MGR did not do review and editing of references.

Author Response

Response to the reviewer 1

Manuscript 921636

It is a lengthy review manuscript in which authors attempted to prove the concept that the real origin of problems encountered in DR starts in photoreceptors, notably by the massive production of ROS. Authors try to prove that vascular problems develop in later stages of the disease and they are preceded by huge mitochondrial abnormalities of outer retinal layers that start to occur in early stages of DR. Personally, I believe it make sense however, it is not absolutely sure that this concept presents the only one possible scenario in this complicated disease.

Response: We thank the reviewer for the valuable suggestions to improve our manuscript. We have acknowledged in our revised version that, indeed, DR is a very complex disease that involves multiple pathogenic mechanisms (Line 15-16, Abstract). We have clarified our argument that as antioxidant therapeutic approaches have proved beneficial in animal models it is reasonable to believe that oxidative stress is an upstream pathogenic mechanism that enhance other mechanistic pathways including inflammation and apoptosis. We also modified Figure 2 in order to emphasize the complexity of this disease and show the link between alterations in mitochondrial bioenergetics with other pathogenic mechanisms of DR.

The manuscript is not so explicitly written, thus, it requires significant corrections before it can be published. Overall, the ms is quite messy.

Response: In response to the reviewer’s suggestions, we have made extensive revision of the manuscript. For the reviewer’s convenience, we marked in red the additions that replace the eliminated text. We have excluded useless words and redundancies.   

What should be done is mainly to shift accents. First part of the manuscript is far too long and although it presents physiology of the retina and pathophysiology of DR quite well, only late section 6 in page 12 is devoted to oxidative stress in the diabetic retina.

Response: Our goal has been to review the alterations in mitochondrial bioenergetics during diabetes. As noted in line 83: “While performing the core metabolic function of energy production, mitochondria are critical gears in a currently expanding number of cellular functions including redox homeostasis and programmed cell death. An increased mitochondrial oxidative stress reveals a change in mitochondrial function. The importance of understanding the role of bioenergetics in the diabetic neural retina is supported by the knowledge that inherited mitochondrial diseases cause retinal disease and visual impairment, and is further highlighted by the heterogeneity of the neural retinal cells regarding the contribution of their mitochondria to cellular ATP and oxidative stress. This review will summarize the recent developments related to alterations in mitochondrial bioenergetics in the neural retina, and the consequences of these alterations on retinal function. We will conclude with therapeutic implications, acknowledging emerging therapeutic approaches to correct mitochondrial bioenergetic-related functions, and the broader future directions of this research focused on maintaining mitochondrial integrity in diabetes.”

Therefore, we start with discussing the normal and unique retinal mitochondrial metabolism, and continue with alterations in multiple mitochondrial functions (oxidative phosphorylation, redox state) as potential causes of the increased mitochondrial ROS generation. We have addressed the complex role of mitochondria in the neural retina and microvascular disease in the diabetic retina (as mentioned in the title as well). However, we have extensively eliminated useless words and redundancies from the first part of the manuscript.

Just to remind you, authors promised in Abstract: " to discuss the recent developments related to alterations in mitochondrial substrate selection and bioenergetics............"  and " emerging therapeutic approaches and future research directions......." Especially this last part is underdeveloped.

Response: We have made extensive changes in our review in order to emphasize that alterations in mitochondrial bioenergetics and substrate selection in diabetes are upstream to increased mitochondrial oxidative stress. We also discuss in the current version of the manuscript novel therapeutic approaches that are targeted to protect mitochondrial integrity and rather decrease mitochondrial ROS generation that scavenging already generated ROS. This discussion starts at line 632 in the current manuscript version). Although we expanded the last part, we were able to shorten the review from 762 lines to 728 lines (approximately 3/4 of a page).

It should be more "cutting edge" otherwise it adds very little to the current knowledge. I would avoid repeated ideas and sentences in which this ms is quite rich.

Response: The strength of our review relies on the focus on alterations in mitochondrial bioenergetics as a complex of functions (maintenance of the redox state, ATP generation) leading to increased oxidative stress. We have made extensive revisions, especially in the first part of the manuscript, in order to eliminate repetitions and redundancies.

References is a misunderstanding, there are lacking authors, titles, abbreviations of the names of journals, volumes etc. It is very messy.

Response: We have used the EndNote program using the ACS guidance for the references style (www.mdpi.com). In addition, we have used both the ISO 4 rules (ISSN Center List of Title Word Abbreviations) and CAS (Division of American Chemical Society) Core Journals List in order to abbreviate the cited journal titles.

MAC and MGR did not do review and editing of references.

Response: All authors have now reviewed and edited the references. A sentence was added in the Author contributions section.

Reviewer 2 Report

The reviewed manuscript is a very interesting summary of the role of mitochondrial dysfunction in diabetic retinopathy. However, the article is definitely too long. It requires data synthesis as well as the collection of data in the form of tables with the endpoints of the studies used in the review. These tables will be the perfect complement to the data collected in the summary tables at the end of the manuscript. The abstract also requires rebuilding, it should summarize the subject of the article more closely.

Author Response

Response to the reviewer 2

Manuscript 921636

The reviewed manuscript is a very interesting summary of the role of mitochondrial dysfunction in diabetic retinopathy. However, the article is definitely too long. It requires data synthesis as well as the collection of data in the form of tables with the endpoints of the studies used in the review. These tables will be the perfect complement to the data collected in the summary tables at the end of the manuscript. The abstract also requires rebuilding, it should summarize the subject of the article more closely.

Response: We are very appreciative for the reviewer’s suggestions to improve our manuscript. We have shortened the first part of the review and focused more on alterations of mitochondrial functions that are potentially responsible for increased oxidative stress. We also expanded on the mitochondrial targeted therapeutic approaches (in red). We have also revised the abstract (in red). Although we expanded the last part, we were able to shorten the review from 762 lines to 728 lines (approximately 3/4 of a page) by eliminating redundancies and useless words.

We have two Tables included in the manuscript, which include the data related to mitochondrial abnormalities (oxidative phosphorylation, specific activities of mitochondrial complexes).

Reviewer 3 Report

This review highlights the role of mitochondria during Diabetic retinopathy.

The review is well written, updated, and deals with an important topic in the field. I enjoyed the clear link between retina function and mitochondria, and the exhaustive introduction to the problem, which makes the review fruible also to a larger audience.

Images are clear and caption either.   

Check for the few errors throughout the text (ex. line 147, 234)

Author Response

Response to the reviewer 3

Manuscript 921636

This review highlights the role of mitochondria during Diabetic retinopathy.

The review is well written, updated, and deals with an important topic in the field. I enjoyed the clear link between retina function and mitochondria, and the exhaustive introduction to the problem, which makes the review fruible also to a larger audience.

Images are clear and caption either.   

Check for the few errors throughout the text (ex. line 147, 234)

Response: We thank this reviewer of appreciating our work, and corrected the errors that were suggested as well as others that we have found in the text. In addition, in response to other reviewers suggestions, we have made revisions of the text that are highlighted in RED in the current version of the manuscript.

Round 2

Reviewer 1 Report

Now manuscript seems even longer

References are still messy (various abbreviations, etc.)

Author Response

Reviewer 1

Now manuscript seems even longer

Response: With all the additions to the second part of the manuscript, including the therapeutic implications, we have shorten the manuscript from 762 lines (first submitted version) to 714 lines (second revised version) due to deletions and eliminations of redundancies from (mainly) the first part, as the reviewer suggested. Therefore, we now have a more balanced review that is also approximately one page shorter compared to the original submission.

References are still messy (various abbreviations, etc.)

Response: We have corrected all the errors in the reference section according to the EndNote program using the ACS guidance for the references style (www.mdpi.com). The corrections are tracked. The doi number is not assigned in PubMed for references 54, 63, 69, 80 and 86.

Reviewer 2 Report

The manuscript has been significantly improved, but still needs to be changed.
Abstract: delete verses 14-15
Line 72: add newer citations, e.g. 10.1155/2019/2606120
Add missing citations: lines 86, 163-175, 191, 195-203, 211, 214-227, 283, etc.
Table 1: explain the meaning of the symbol.
Table 2: explain the meaning of the arrows.
